An account of Korthalsia (Arecaceae) rattans and their uses in Thailand

Ngernsaengsaruay Chatchai 1 2 fsciccn@ku.ac.th
Mianmit Nittaya 3 ffornym@ku.ac.th
Leksungnoen Nisa 4
Racharak Phruet 4
http://orcid.org/0000-0003-2459-8564 Uthairatsamee Suwimon 4
http://orcid.org/0009-0001-7325-6109 Chanton Pichet 5
Andriyas Tushar 6
Duangjai Wirongrong 7
1 Department of Botany, Faculty of Science, Kasetsart University , Chatuchak, Bangkok , Thailand
2 Biodiversity Center Kasetsart University (BDCKU), Kasetsart University , Chatuchak, Bangkok , Thailand
3 Department of Forest Management, Faculty of Forestry, Kasetsart University , Chatuchak, Bangkok , Thailand
4 Department of Forest Biology, Faculty of Forestry, Kasetsart University , Chatuchak, Bangkok , Thailand
5 Suan Luang Rama IX Foundation , Nong Bon Subdistrict, Prawet District, Bangkok , Thailand
6 Department of Food and Pharmaceutical Chemistry, Faculty of Pharmaceutical Sciences, Chulalongkorn University , Pathumwan, Bangkok , Thailand
7 Department of Silviculture, Faculty of Forestry, Kasetsart University , Chatuchak, Bangkok , Thailand
Sosa Victoria
Electronic publication date: 2025 Aug 19
Publication date: 2025
Volume: 13
Electronic Location ID: e19935
Received 2025 May 6; Accepted 2025 Jul 25
Copyright: © 2025 Ngernsaengsaruay et al.
Copyright year: 2025
Copyright holder: Ngernsaengsaruay et al.
License: This is an open access article distributed under the terms of the Creative Commons Attribution License, which permits unrestricted use, distribution, reproduction and adaptation in any medium and for any purpose provided that it is properly attributed. For attribution, the original author(s), title, publication source (PeerJ) and either DOI or URL of the article must be cited.
License URL: https://creativecommons.org/licenses/by/4.0/

Keywords: Calamoideae, Climbing palms, Ethnobotany, Hapaxantic palms, Hermaphroditic palms, Korthalsiinae, Ocrea, Palmae, Rhomboid leaflets, Taxonomy

Funding: Science, Research and Innovation Promotion Fund of the Thailand Science Research and Innovation (TSRI) Kasetsart University Research and Development Institute (KURDI) FF(KU)41.68 This work was financially supported by Science, Research and Innovation Promotion Fund of the Thailand Science Research and Innovation (TSRI) through Kasetsart University Research and Development Institute (KURDI), FF(KU)41.68. The funders had no role in study design, data collection and analysis, decision to publish, or preparation of the manuscript.

==============================
Four species of Korthalsia are found in Thailand: K. flagellaris, K. laciniosa, K. rigida, and K. scortechinii. We present the comparative vegetative morphology, a key to the species, distribution, ecology, preliminary conservation assessments, utilization, and specimens examined. Korthalsia flagellaris is lectotypified here in a second-step. Three Thai Korthalsia species are confined to the peninsular region: K. flagellaris, K. scortechinii and K. rigida. The first, K. flagellaris, is known only from Narathiwat province and is restricted to peat swamp forests. The second, K. scortechinii, is myrmecophilous and found only in Narathiwat in lowland tropical evergreen rainforests. The third, K. rigida has a wider distribution and is known from Ranong, Trang and Narathiwat provinces in lowland tropical evergreen rainforests. Lastly, Korthalsia laciniosa is the most widespread species of the genus in both the south-eastern and peninsular regions of Thailand in lowland tropical evergreen rainforests. All Thai Korthalsia species are currently assessed as Least Concern (LC) under the IUCN conservation status. In Thailand, Korthalsia is used for traditional tools, furniture components, and fishery equipment. Its stems are valued in local crafts, while the young shoots are occasionally consumed as food.

Introduction

Korthalsia Blume is one of the eight genera of rattans (POWO, 2025; WFO, 2025) which are characterized by clustered, spiny, high-climbing, and aerially branching growth habits, They are hapaxanthic and hermaphroditic (Dransfield et al., 2008; Barfod & Dransfield, 2013). Morphologically, Korthalsia is a very distinctive and well-circumscribed genus, differing markedly from other rattan genera in many features. Most species exhibit greater variation in vegetative characteristics than in inflorescence details, allowing identification of sterile material (Dransfield, 1981). The genus name Korthalsia was named by Carl (Karl) Ludwig Blume (1796–1862), a German-Dutch botanist who worked extensively in Java and served as director of the Rijksherbarium at Leiden (Blume, 1843; Stafleu & Cowan, 1976). The name honors Pieter Willem Korthals (1807–1892), a Dutch botanist and explorer in the East Indies, who first collected specimens in Indonesia (Stafleu & Cowan, 1979; Riffle & Craft, 2003). The common name for Korthalsia is ant-rattan (Dransfield et al., 2008).

Korthalsia is the sole genus in the subtribe Korthalsiinae Becc. within the tribe Calameae Kunth ex Lecoq. & Juillet, subfamily Calamoideae Griff., family Arecaceae Bercht. & J. Presl (Palmae Juss., conserved name), and order Arecales Bromhead (Dransfield et al., 2008).

Korthalsiinae are climbing, hapaxanthic, hermaphroditic palms with stems that are often aerially branching. Leaves possess cirri but lack acanthophylls (large spines derived from leaflets), and leaflet apices are praemose. The inflorescence is adnate to the internode, and the seeds lack a sarcotesta (Dransfield et al., 2008). The genus Korthalsia is distributed from Indo-China, Myanmar, and the Andaman Islands south-eastwards to Sulawesi and New Guinea (Dransfield et al., 2008). Of the 27 accepted Korthalsia species (POWO, 2025), 10 are myrmecophilous (Shahimi et al., 2019). The genus is widespread throughout the Malesian region, with most species found in Borneo (15), Peninsular Malaysia (nine), and Sumatra (nine) (POWO, 2025; WFO, 2025). All species are restricted to lowland and hill tropical rainforests, and are absent from montane forests (Dransfield et al., 2008).

Korthalsia species provide valuable canes used for making baskets, mats, fish-traps, furniture, and various handicrafts. They are also used as binding materials in construction. Some species are employed in traditional medicine or consumed as food, while others are cultivated as ornamentals or used in floral arrangements (Dransfield, 1981; Baja-Lapis, 2009; Shahimi et al., 2019; Barfod & Dransfield, 2013; Senthilkumar et al., 2014; Dasgupta et al., 2021; Nugroho et al., 2022).

Species of Korthalsia produce long, hard, and durable canes that are widely used in local basketry and for binding in house construction (Dransfield, 1981; Dransfield et al., 2008). Large-diameter canes are produced by only a few species, notably Korthalsia ferox, K. laciniosa, and K. flagellaris, and are commonly used in the construction of large sea fish-traps and occasionally for the frameworks of inexpensive armchairs (Dransfield, 1981). Smaller-diameter canes are used whole or split for binding and weaving. Some of the most durable and aesthetically appealing carrying baskets in Southeast Asia are made from Korthalsia canes (Dransfield, 1981).

However, the canes are often disfigured by large, irregular nodal scars, and the inner epidermis of the sheaths tends to adhere tightly to the cane surface. These characteristics have limited the commercial value of Korthalsia in the rattan trade (Dransfield, 1981; Dransfield et al., 2008). Additionally, the fruits of most species appear to be attractive to animals due to a sweet, thin, fleshy layer surrounding the seed (Dransfield, 1981).

A revision of the genus Korthalsia in Thailand was published by Barfod & Dransfield (2013), in which five species were recognised: K. flagellaris Miq., K. laciniosa (Griff.) Mart., K. rigida Blume, K. rostrata Blume, and K. scortechinii Becc. K. rostrata was included in the “The Palms and Cycads of Thailand” and reported from Narathiwat province (Hodel & Vatcharakorn, 1998). However, we have found no material from Thailand that matches the descriptions of K. rostrata as defined by Barfod & Dransfield (2013). Korthalsia rostrata is one of the ten known myrmecophilous species and is distributed in Peninsular Malaysia, Singapore, Sumatra, and Borneo (Brunei, Sabah, Sarawak, Central, East, and West Kalimantan) (Shahimi et al., 2019), but it has not been confirmed as part of the Thai flora.

Here, we present an updated comparative account of the vegetative morphology, distribution, ecology, conservation status, and traditional uses of the four Korthalsia species recorded in Thailand: K. flagellaris, K. laciniosa, K. rigida, and K. scortechinii. This study forms part of the research project entitled “The sustainable research and development of rattan: a case study of the genus Korthalsia”. Our findings serve as a foundation for advancing research in the taxonomy, ecology, ethnobotany, and conservation of Korthalsia and related palms. They also provide a basis for future research and development in related scientific disciplines.

Materials and Methods

Korthalsia species were observed and specimens were collected in the south-eastern and peninsular regions of Thailand. The collected specimens were examined and compared to the literature (e.g., Whitmore, 1977; Dransfield, 1981; Tomlinson, 1990; Barfod & Dransfield, 2013; Shahimi, 2018; Shahimi et al., 2019) and with herbarium specimens housed in the following herbaria: BK, BKF, and those included in the digital herbarium databases of A (including GH), AAU, BR, E, GH, K, L (including U), P, SING, US, and MICH, MO, MW, and NY (via GBIF, https://www.gbif.org/). All herbarium codes follow Thiers (2025). Vegetative morphological features, distributions, habitats and ecology, and uses were described from historical and newly collected herbarium specimens, as well as the authors’s field observations. Vernacular names were compiled from specimens examined and relevant literature (e.g., Dransfield, Barfod & Pongsattayapipat, 2004; Barfod & Dransfield, 2013; Pooma & Suddee, 2014). Thailand floristic regions follow Flora of Thailand Vol 4(3.3) (The Forest Herbarium, Department of National Parks & Wildlife and Plant Conservation, 2023). A preliminary assessment of conservation status was conducted following the International Union for Conservation of Nature (IUCN) Red List Categories and Criteria (IUCN Standards and Petitions Committee, 2024), supplemented with GeoCAT analysis (Bachman et al., 2011) and field data. Leaf shape (length/width ratio) was assessed following Simpson (2010). The methods used in this study are primarily based on Ngernsaengsaruay et al. (2024, 2025). Permission to collect specimens was obtained from the Department of National Parks, Wildlife and Plant Conservation, Ministry of Natural Resources and Environment (MNRE 0910.5803/7512).

Results

Vegetative Morphology

Habit

Korthalsia is a genus of slender, moderate-sized to robust, clustering, high-climbing rattan palms up to 50 m or more long. Many species can reach the forest canopy. All Korthalsia species are hapaxanthic (individual stems produce flowers only once in their lifetime and die subsequently). Thai Korthalsia species can be myrmecophilous, i.e., living in close association with ants, K. scortechinii or non-myrmecophilous, K. flagellaris, K. laciniosa, and K. rigida.

Stems

Korthalsia is the only genus of Southeast Asian and Malesian rattans which frequently branch in the forest canopy (aerially branching), or sometimes basally branching (in K. rigida). The mature stems are hard and durable. The nodes are often uneven or marked with scars of branches (nodal scars). The lowest node of the stem bears adventitious roots (as observed in K. laciniosa). The internodes are elongate and variable in length. The stem diameter include leaf sheaths ranges from 1.4–5.2 cm diam. (in K. laciniosa, 3–8 cm diam. at the base).

Leaves

The leaves of Korthalsia are once-pinnately compound, spreading, and 1–4 m long including petiole and cirrus. The number of leaflets on each side of the rachis varies from 5–21 and they are regularly arranged, rather distant. The rachis is extended into a slender long cirrus (ranges from 40–180 cm long), and armed with scattered single, sometimes paired or grouped, and recurved spines (hooked spines) as the climbing organ. The petiole is from 8–52 cm long, and armed with sparsely single, paired or grouped and recurved or erect spines. The rachis is from 40–210 cm, and sparsely armed with single, paired or grouped and recurved spines.

The leaf sheaths are tubular, sometimes splitting longitudinally opposite the petiole, green, pale yellowish-green, dull green, dull greenish-pale brown or dull pale brown and they are usually with caducous scales, or sometimes densely covered with persistent whitish-grey indumentum (in K. rigida) and are variously armed with single, paired or grouped and flattened triangular or bulbous-based (in K. rigida) spines, or sometimes unarmed (in K. flagellaris). The inner epidermis of the leaf sheaths tends to adhere to the stem surface. The leaf sheaths are without a knee (a swelling on the leaf sheath at the base of the petiole).

All species of Korthalsia have a conspicuous extension of the leaf sheath beyond the petiole insertion known as ocrea. The ocrea types of Thai Korthalsia species include tightly sheathing (in K. flagellaris, K. laciniosa, and K. rigida) or inflated and forming an ant chamber (in K. scortechinii), with scales as the leaf sheath, and are variously armed with single, paired or grouped, flattened triangular or bulbous-based (in K. rigida) and erect or slightly recurved spines, or sometimes unarmed (in K. flagellaris).

Leaflets

The leaflets of Korthalsia are alternate, sometimes subopposite or opposite, and usually plicate. The shape of the leaflets is usually rhomboid (diamond-shaped), sometimes broadly, narrowly or very narrowly rhomboid with the distal margins praemorse and their size is 12–40 × 1.5–22 cm. The proximal leaflets are often smaller than the distal ones (except in K. rigida). The leaflet length/width ratio ranges from 1.2–17.8. The adaxial surface of the leaflets is usually glossy dark green and glabrous (except in K. rigida, which has main veins with black and reddish-dark brown scales), and the abaxial surface is glaucous (except in K. flagellaris) and covered with black, reddish-dark brown or reddish-brown scales. The venation is divergent parallel with 5–17 main veins gradually spreading from the leaflet base. Transverse veinlets are conspicuous on the adaxial surface of leaflets. The petiolule is absent (in K. scortechinii) to well-developed and is laterally flattened with black, blackish-brown, reddish-brown or reddish-dark brown scales (in K. rigida with persistent whitish-grey indumentum). The petiolule length ranges from 0.4–6 cm.

A key to the species of Korthalsia in Thailand

1a. Myrmecophilous climbing rattan; ocrea inflated and forming an ant chamber, ellipsoid, 9–19.5 × 5.7–10 cm, and armed with scattered short, flattened triangular and erect or recurved spines, 1.5–6 mm long; petiolule absent; leaflets narrowly rhomboid or rarely rhomboid [length/width ratio (2.6–)3–6.3]4. Korthalsia scortechinii

1b. Non-myrmecophilous climbing rattan; ocrea tightly sheathing; petiolule present; leaflets rhomboid or very narrowly rhomboid, sometimes narrowly or broadly rhomboid……………………2

2a. Leaflets very narrowly rhomboid or sometimes narrowly rhomboid, the widest ones up to 5 cm [length/width ratio (4.6–)6–17.8], not glaucous, covered with densely reddish-brown and black scales on abaxial surface, with 5–8 main veins, with 13–21 leaflets on each side of rachis; restricted to peat swamp forest and known only in Narathiwat province…………………………1. Korthalsia flagellaris

2b. Leaflets rhomboid, sometimes narrowly or broadly rhomboid, the widest ones more than 5 cm [length/width ratio (1.2–)1.5–3(–4.4)], glaucous on abaxial surface, with (5–)7–17 main veins, with 5–15 leaflets on each side of rachis; found in lowland tropical evergreen rainforests………………………………3

3a. Ocreas 15–32.5 cm long, pale brown or reddish-brown, tending to erode into a loosely fibrous net-like tube; leaf sheath green or sometimes pale yellowish-green, glabrous, sparsely armed with flattened triangular spines; distributed in the south-eastern (Chanthaburi and Trat provinces) and the peninsular (Chumphon, Surat Thani, Krabi, Nakhon Si Thammarat, Phatthalung, Trang, Satun, Songkhla, Yala, and Narathiwat provinces) regions2. Korthalsia laciniosa

3b. Ocreas 3.5–7.5 cm long, dull green with a reddish-brown or reddish-dark brown margin, apex truncate with a narrow tattering margin, sometimes splitting or tending to erode into a fibrous net-like; leaf sheath dull green, densely covered with persistent whitish-grey indumentum, armed with scattered flattened triangular or bulbous-based spines; known only in the peninsular region (Ranong, Trang, and Narathiwat provinces)…………………………………………………………………………3. Korthalsia rigida

1. Korthalsia flagellaris Miq., Fl. Ned. Ind., Eerste Bijv. Suppl. Pt. 3: 591. 1861 (Miquel, 1861). Type: Indonesia, Sumatra, Priaman, s.d., H. Diepenhorst s.n. (Herb. Bogor 2584) (lectotype, first-step designated by Dransfield (1981), U [without barcode]; isolectotype FI [without barcode], second-step designated here U digital image! [U1140217]; isolectotypes U digital image! [U1140216], FI digital image! [FI086077]).

= Korthalsia rubiginosa Becc., Malesia 2: 72. 1884. Type: Malaysia, Borneo, Sarawak, G. Matang, Jun 1866, O. Beccari (Piante Bornensi 1912) (holotype FI digital image! [FI013516]).

Non-myrmecophilous, robust, clustered, high-climbing rattan, up to 40 m or more long. Stems including leaf sheaths 1.6–4.7 cm diam.; internodes 20–40 cm long, cut stems with scarce clear exudate. Leaves 2.16–2.92 m long including petiole and cirrus; leaflets 13–21 on each side of rachis, regularly arranged; leaf sheath dull greenish-pale brown, dull pale brown, or sometimes green, covered with caducous black and pale brown scales, and usually unarmed; ocrea 8–19 cm long, reddish-brown or reddish-dark brown, closely sheathing at first but tending to dissolve into a loosely fibrous net-like tube, with scales as the leaf sheath, and usually unarmed; petiole 14–27 cm long, 0.9–1.7 cm wide and 4.5–8.5 mm thick, adaxially shallowly grooved or flattened proximally and flattened distally, abaxially rounded, petiole 17.5–42 cm apart from each other, sparsely armed with short, single or sometimes paired and recurved or erect spines, each spine 2.5–8 mm long; rachis 0.91–1.22 m long, adaxially flattened proximally and angular distally, sparsely armed with short, single, paired or grouped and recurved spines, each spine 2.5–9 mm long; petiole and rachis green adaxially, pale green or pale yellowish-green abaxially, and with scales as the leaf sheath; cirrus slender, green, 0.91–1.57 m long, armed with scattered short, grouped, sometimes paired or single and recurved spines. Leaflets alternate, very narrowly rhomboid or sometimes narrowly rhomboid, 18.5–37 × 1.5–5 cm [length/width ratio (4.6–)6–17.8], the lowest pair of leaflets often smaller than the others, the distal margins praemorse, adaxially glossy dark green and glabrous, abaxially paler, not glaucous and densely covered with black and reddish-brown scales proximally, with densely reddish-brown scales distally, main veins 5–8, diverging from the lamina base, raised above, transverse veinlets conspicuous above; petiolule laterally flattened, 0.4–2 cm long, with caducous black scales; young leaflets pale green (Fig. 1).

Figure 1 Korthalsia flagellaris.

(A) Habitat. (B) Habit, showing leaves terminating in a long cirrus. (C) Tightly sheathing ocrea clasping the stem, closely sheathing at first but tending to erode into a loosely fibrous net-like tube. (D) Leaf and leaflets on adaxial surface. (E) Leaf and leaflets on abaxial surface, covered with densely reddish-brown scales. (F) Cut stem and leaflets on both surfaces. (A–F) from Sirindhorn Peat Swamp Forest Nature Research and Study Center (To Daeng Peat Swamp Forest), Narathiwat Province. Photos: Chatchai Ngernsaengsaruay.

Distinguishing characteristics. Korthalsia flagellaris can be recognised as a robust high-climbing rattan; leaves with 13–21 leaflets on each side of rachis; shape and size of leaflets, very narrowly rhomboid or sometimes narrowly rhomboid and 18.5–37 × 1.5–5 cm [length/width ratio (4.6–)6–17.8]; leaflets not glaucous and covered with densely reddish-brown and black scales on abaxial surface; ocreas tightly sheathing, closely sheathing at first but tending to dissolve into a loosely fibrous net-like tube; and unarmed leaf sheaths and ocreas.

Distribution, ecology and conservation status. Korthalsia flagellaris is widely distributed from Peninsular Thailand to Sumatra and Borneo (Fig. 2A). It is known from numerous localities with a large Extent of Occurrence (EOO) estimated at 1,629,571.58 km2. However, its Area of Occupancy (AOO) is relatively small, at only 100 km2. In Thailand, this species is known exclusively from Narathiwat province, where it is restricted to peat swamp forests, at elevations from near sea level to approximately 30 m a.s.l. Within Thailand, it has a much smaller EOO of 109.056 km2 and AOO of 12 km2. Despite being a widespread species regionally, K. flagellaris is threatened in Thailand due to the destruction of its limited and specialized habitat (Barfod & Dransfield, 2013). Nonetheless, given its broad overall distribution and occurrence across multiple localities, we propose a conservation assessment as of Least Concern (LC) under the IUCN Red list criteria.

Figure 2 Distribution of Thai Korthalsia species.

(A) Korthalsia flagellaris is widely distributed from Peninsular Thailand to Sumatra and Borneo. In Thailand, this species is known only from Narathiwat Province (B) Korthalsia laciniosa is the most widespread species of the genus and is distributed from Andaman and Nicobar Islands, Southern Myanmar, Indo-China to Philippines. In Thailand, this species is distributed in the south-eastern and the peninsular regions (C) Korthalsia rigida is widely distributed from Southern Myanmar, Peninsular Thailand to Borneo and Philippines (Palawan). In Thailand, this species is known only in the peninsular region (Ranong, Trang and Narathiwat Provinces) (D) Korthalsia scortechinii is a species with a narrow distribution from Peninsular Thailand to Peninsular Malaysia. In Thailand, this species is known only in Narathiwat Province. Maps: Pichet Chonton and Chatchai Ngernsaengsaruay.

Vernacular names. Wai dao nam (หวายเดาน้ำ), Wai sadao nam (หวายสะเดาน้ำ) (Narathiwat); Ro-tan-da-nae (รอตันดาแน) (Malay-Narathiwat, from the specimens C. Niyomdham 574 and C. Niyomdham 826).

Uses. In Narathiwat province, the young shoots of Korthalsia flagellaris are eaten even if they have a distinctly bitter taste. They are typically boiled and eaten with chili paste. This species produces a coarse cane (Barfod & Dransfield, 2013).

Distribution and utilization of the rattan genus Korthalsia in Thailand and other countries is shown in Table 1.

Table 1 Distribution and utilization of the rattan genus Korthalsia in Thailand and other countries.

Species	Distribution	Utilization	
*1. Korthalsia angustifolia Blume	Borneo [Indonesia (Central and South Kalimantan)]	Not recorded	
2. Korthalsia bejaudii Gagnep. ex Humbert	Cambodia	Not recorded	
3. Korthalsia celebica Becc.	Indonesia [Sulawesi, Maluku (also called Moluccas)]	Not recorded	
*4. Korthalsia cheb Becc.	Borneo [Malaysia (Sabah, Sarawak), Indonesia (East and South Kalimantan)]	The stems of Korthalsia cheb are used for making baskets and as binding material for constructing pig-sties (Shahimi et al., 2019).	
5. Korthalsia concolor Burret	Borneo	Not recorded	
6. Korthalsia debilis Blume	Indonesia (Sumatra), Borneo	Not recorded	
*7. Korthalsia echinometra Becc.	Peninsular Malaysia [Perak, Terengganu, Pahang, Johor (also spelled Johore)], Singapore, Indonesia (Sumatra), Borneo [Brunei, Malaysia (Sabah, Sarawak), Indonesia (Kalimantan)]	The canes of Korthalsia echinometra are used to make basket frames, weaving handicrafts and also to tie planks on dugouts to raise the side wall of canoes. Sap can be drunk for fever (Shahimi et al., 2019).	
8. Korthalsia ferox Becc.	Borneo	The large diameter canes of Korthalsia ferox are used in the construction of large fish-traps out at sea, and occasionally for the framework of cheap armchairs (Dransfield, 1981).	
		The stems of Korthalsia ferox are used in the treatment of heartburn or stomach pain and the shoots are cleaned and then eaten directly (Nugroho et al., 2022).	
**9. Korthalsia flagellaris Miq.	Peninsular Thailand, Peninsular Malaysia (Pahang, Selangor), Singapore, Indonesia (Sumatra), Borneo [Brunei, Malaysia (Sabah, Sarawak), Indonesia (East, Central and South Kalimantan)]
Distribution in Thailand. Peninsular: Narathiwat (Fig. 2A)	The large diameter canes of Korthalsia flagellaris are used in the construction of large fish-traps out at sea, and occasionally for the framework of cheap armchairs (Dransfield, 1981). Produces a coarse cane (Barfod & Dransfield, 2013).	
*10. Korthalsia furcata Becc.	Borneo [Malaysia (Sarawak), Indonesia (West Kalimantan)]	Not recorded	
*11. Korthalsia furtadoana J. Dransf.	Borneo [Brunei, Malaysia (Sabah), Indonesia (Central, East and South Kalimantan)]	Not recorded	
*12. Korthalsia hispida Becc.	Peninsular Malaysia (Pahang, Johor), Indonesia (Sumatra), Borneo [Brunei, Malaysia (Sabah, Sarawak), Indonesia (East and South Kalimantan)]	Fibres of Korthalsia hispida are used for the plaiting of baskets, craft and binding constructions (Shahimi et al., 2019).	
13. Korthalsia jala J. Dransf.	Borneo	Not recorded	
14. Korthalsia junghuhnii Miq.	Indonesia (Java)	Not recorded	
**15. Korthalsia laciniosa (Griff.) Mart.	Andaman and Nicobar Islands, Southern Myanmar (Mergui Archipelago), Indo-China (Vietnam, Laos, Cambodia), Peninsular Thailand, Peninsular Malaysia (Kedah, Perak, Terengganu, Pahang, Selangor, Negeri Sembilan, Johor), Singapore, Indonesia (Sumatra, Java), Philippines (Luzon, Palawan, Mindanao)
Distribution in Thailand. South-Eastern: Chanthaburi, Trat; Peninsular: Chumphon, Surat Thani, Krabi, Nakhon Si Thammarat, Phatthalung, Trang, Satun, Songkhla, Yala, Narathiwat (Fig. 2B)	The large diameter canes of Korthalsia laciniosa are used in the construction of large fish-traps out at sea, and occasionally for the framework of cheap armchairs (Dransfield, 1981). It is a source of durable and flexible cane and used in making fence, rafts and as an ornamental species (Baja-Lapis, 2009; Senthilkumar et al., 2014; Dasgupta et al., 2021). The species has a high market demand, but is exported in limited quantity (3%) due to significant local demand (Senthilkumar et al., 2014). K. Rubeli observed Southern Pied Hornbills (Anthracoceros convexus) feeding on the ripe fruits of Korthalsia laciniosa in Taman Negara, Pahang, Peninsular Malaysia in 1978 (Dransfield, 1981, personal communication). Produces a coarse cane (Barfod & Dransfield, 2013). Young K. laciniosa seedlings are used as indoor plants and ornamental garden plants. The leaves are used in flower arrangements (Baja-Lapis, 2009).	
16. Korthalsia lanceolata J. Dransf.	Peninsular Malaysia	Not recorded	
17. Korthalsia merrillii Becc.	Philippines	Not recorded	
18. Korthalsia minor A. J. Hend. & N. Q. Dung	Vietnam, Laos	Not recorded	
19. Korthalsia paucijuga Becc.	Indonesia (Sumatra), Borneo	Not recorded	
**20. Korthalsia rigida Blume	Southern Myanmar (Mergui Archipelago), Peninsular Thailand, Peninsular Malaysia [Perak, Kelantan, Pahang, Selangor, Negeri Sembilan, Maleka (also spelled Malacca), Johor], Singapore, Indonesia (Sumatra), Borneo [Brunei, Malaysia (Sabah, Sarawak), Indonesia (North, East, Central, and South Kalimantan)], Philippines (Palawan)
Distribution in Thailand. Peninsular: Ranong, Trang, Narathiwat (Fig. 2C)	Produces a coarse cane (Barfod & Dransfield, 2013).	
*21. Korthalsia robusta Blume	Indonesia (Sumatra), Borneo [Malaysia (Sabah, Sarawak), Indonesia (East, Central, and South Kalimantan)], Philippines (Palawan)	The stems of Korthalsia robusta are used for tying and weaving of baskets, handicraft and construction and the shoots are edible (Shahimi et al., 2019).	
22. Korthalsia rogersii Becc.	Andaman Islands	The stems of Korthalsia rogersii are use in handicraft industries and the leaves are used for decorative purposes (Senthilkumar et al., 2014).	
*23. Korthalsia rostrata Blume	Peninsular Malaysia (Perak, Terengganu, Pahang, Selangor, Melaka, Johor), Singapore, Indonesia (Sumatra), Borneo [Brunei, Malaysia (Sabah, Sarawak), Indonesia (Central, East and West Kalimantan)]	The stems of Korthalsia rostrata are used for weaving basket and mats (Shahimi et al., 2019). The slender canes are used in some basketry and for tying (Barfod & Dransfield, 2013).	
*24. Korthalsia scaphigeroides Becc.	Philippines (Mindanao)	The stems of Korthalsia scaphigeroides are used to make furniture and basket (Shahimi et al., 2019).	
*/**25. Korthalsia scortechinii Becc.	Peninsular Thailand, Peninsular Malaysia (Kedah, Penang, Perak, Pahang, Negeri Sembilan, Johor)
Distribution in Thailand. Peninsular: Narathiwat (Fig. 2D)	The canes of Korthalsia scortechinii are used to make baskets, also as binding material (Shahimi et al., 2019).	
26. Korthalsia tenuissima Becc.	Peninsular Malaysia	Not recorded	
27. Korthalsia zippelii Blume	New Guinea, Bismarck Archipelago	Not recorded	
Notes:

* Myrmecophilous rattans.

** Korthalsia species found in Thailand.

Notes. Korthalsia flagellaris was originally described by Miquel (1861), who cited material collected by Diepenhorst from Sumatra, in the province of Priaman. However, Miquel did not select a holotype nor did he mention the herbarium in which the material was deposited. Later, Dransfield (1981) selected the same material (Herb. Bogor 2584) housed at U [without barcode] as a holotype, with an isolectotype at FI [without barcode]. According to Art. 9.6 of the ICN (Turland et al., 2018), these specimens should be treated as syntypes, and Dransfield’s selection thus constitutes a first-step lectotypification. We have located three sheets of the original material: two at U [U1140216, U1140217] and one at FI [FI086077], all corresponding to Diepenhorst s.n. (Herb. Bogor 2584). Among them, the sheet U1140217 is in the best condition and is hereby designated as the second-step lectotype. The remaining sheets, U1140216 and FI086077, the became isolectotypes.

Korthalsia rubiginosa was described by Beccari (1884), based on material collected from “Sul Monte Mattdin, Sarawak, Borneo (P. B. no 1912)”. Although Beccari did not explicitly designate a holotype or indicate the herbarium in which the specimen was housed, the single gathering cited qualifies as the holotype, in accordance with Art. 9.5 of the ICN (Turland et al., 2018). Dransfield (1981) referred to a specimen at FI [without barcode] as the holotype, consistent with this interpretation. We have located the holotype at FI, represented by digital image FI013516.

Seedlings of Korthalsia flagellaris, have large, simple leaves that resemble young plants of Johannesteijsmannia altifrons (Rchb. f. & Zoll.) H. E. Moore, which typically have entire elliptic or lanceolate leaves, up to 1 m × 15 cm (Dransfield, 1981; Hodel & Vatcharakorn, 1998). However, according to our observations, the leaf shape in K. flagellaris seedlings is oblanceolate.

According to Barfod & Dransfield (2013), Korthalsia flagellaris is distributed in Peninsular Thailand, Peninsular Malaysia, Sumatra, and Borneo. Based on our examinations of herbarium specimens, we have identified two specimens, G. Rajasegar & A. Loo 5 at K [K000667475] and S. Shahimi & A. Loo 20 at K [K001193051] from Singapore’s in the Nee Soon swamp forest which represent an extension of the distribution range and is new record of K. flagellaris in Singapore.

Specimens examined. Thailand. Peninsular: Narathiwat [Sirindhorn peat swamp forest (To Daeng peat swamp forest), Su-ngai Kolok district, sterile, 28 Nov 1990 (as Calamus sp.), A. S. Barfod & W. Ueachirakan 41770 (AAU, BKF); Ban Bang Toei, Tak Bai district, sterile, 2 Jul 1983 (as Korthalsia grandis), C. Niyomdham 574 (BKF); Tak Bai district, with infl., 30 Jul 1984 (as K. grandis), C. Niyomdham 826 (AAU, BKF, K [K000667726], P [P02195943]; Sirindhorn Peat Swamp Forest Nature Research and Study Center (To Daeng peat swamp forest), Puyo subdistrict, Su-ngai Kolok district, 06° 04′ 31″ N, 101° 58′ 03″ E, 30 m, sterile, 6 Feb 2025, C. Ngernsaengsaruay et al. Kf01-06022025 (BKF); ibid., 06° 04′ 28″ N, 101° 58′ 01″ E, 30 m, sterile, 6 Feb 2025, C. Ngernsaengsaruay et al. Kf02-06022025 (BKF); ibid., 06° 04’ 24″ N, 101° 57’ 59″ E, 30 m, sterile, 6 Feb 2025, C. Ngernsaengsaruay et al. Kf03-06022025 (BKF)].

2. Korthalsia laciniosa (Griff.) Mart., Hist. Nat. Palm. 3(7, ed. 2): 211. 1845 (Von Martius, 1845).

≡ Calamosagus laciniosus Griff., Calcutta J. Nat. Hist. 5(17): 23. 1845. Type: Myanmar, Mergui, 1835, W. Griffith s.n. (lectotype designated by Dransfield (1981), BR not seen; isolectotypes FI not seen, K digital image! [K000697940]).

= Calamosagus wallichiifolius Griff., Calcutta J. Nat. Hist. 5(17): 24. 1845 (as C. wallichiaefolius). ≡ Korthalsia wallichiifolia (Griff.) H. Wendl. in O. C. E. de Kerchove de Denterghem, Palmiers: 248. 1878 (as K. wallichiaefolia). Type: Peninsular Malaysia, Malacca, Kussan, s.d., W. Griffith s.n. (holotytpe BR not seen). = Calamosagus harinifolius Griff., Palms Brit. E. Ind.: 29. 1850. Type: as for C. wallichiifolius Griff.

= Korthalsia teysmannii Miq., Fl. Ned. Ind., Eerste Bijv. 3: 591. 1861. Type: Indonesia, Sumatra, Palembang, Muara Dua, s.d., J. E. Teijsmann (Teysmann) s.n (Herb. Bogor 3593)

(lectotype designated by Dransfield (1981), BO not seen; isolectotype FI digital image! [FI086076]).

= Korthalsia scaphigera sensu Kurz, J. Asiat. Soc. Bengal 43(2): 207. 1874 et Forest Fl. Burma 2: 513. 1877 (following Beccari (1884) [non Korthalsia scaphigera Mart., Hist. Nat. Palm. 3(7, ed. 2): 211. 1845], nom. illeg.

= Korthalsia andamanensis Becc., Malesia 2: 76. 1884 (based on K. scaphigera sensu Kurz).

= Korthalsia grandis Ridl., Mat. Fl. Malay. Penins. 2: 217. 1907. Type: Singapore, Seletar, 1894, H. N. Ridley s.n. (lectotype designated by Furtado (1951), SING digital image! [SING0016962]).

Non-myrmecophilous, robust, clustered, high-climbing rattan, up to 50 m or more long. Stems including leaf sheaths 2–5.2 cm diam., 3–8 cm diam. at the base; internodes 20–40 cm long, the lowest node of the stem bearing adventitious roots. Leaves 1.97–3.82 m long including petiole and cirrus; leaflets 6–15 on each side of rachis, regularly arranged and rather distant; leaf sheath green or sometimes pale yellowish-green, covered with caducous reddish-dark brown and black scales, and sparsely armed with single or sometimes paired and flattened triangular spines, each spine 0.4–2.3 cm long, 1.5–5.5 mm wide at the base; ocrea 15–32.5 cm long, pale brown or reddish-brown, closely sheathing at first but tending to dissolve into a loosely fibrous net-like tube, with scales as the leaf sheath, and armed with scattered single or sometimes paired and flattened triangular spines, each spine 0.9–3.2 cm long; petiole 8.5–52 cm long, 0.7–2.1 cm wide and 0.4–1 cm thick, adaxially shallowly grooved or flattened at the lower part and flattened at the upper part, abaxially rounded, petiole 18–48 cm apart from each other, sparsely armed with single or sometimes paired and erect or recurved spines, each spine 3–15 mm long; rachis 0.93–2.08 m long, adaxially flattened proximally and angular distally, abaxially rounded, sparsely armed with single or paired and recurved spines, each spine 4–13 mm long; petiole and rachis green adaxially, pale green or pale yellowish-green abaxially, with scales as the leaf sheath; cirrus slender, green, 0.50–1.80 m long, armed with scattered short, grouped, sometimes paired or single and recurved spines. Leaflets alternate, variable in shape, rhomboid, sometimes narrowly or broadly rhomboid, 15–40 × 5.5–24.8 cm [length/width ratio (1.2–)1.5–3(–4.4)], the lowest pair of leaflets often smaller than the others, the distal margins praemorse, glossy dark green and glabrous on adaxial surface, glaucous and covered with black and reddish-dark brown scales on abaxial surface, or with densely reddish-brown scales at the lower part on both surfaces (especially when young), main veins (5–)7–17, diverging from the lamina base, raised above, transverse veinlets conspicuous above; petiolule laterally flattened, 1–6 cm long, with caducous reddish-brown or reddish-dark brown and black scales; young leaflets pale green (Figs. 3, 4).

Figure 3 Korthalsia laciniosa.

(A–C) Habitats and habit, showing leaves terminating in a long cirrus. (D) The lowest node of the stem bearing adventitious roots. (E and F) Tightly sheathing ocreas clasping the stem, closely sheathing at first but tending to erode into a loosely fibrous net-like tube. (A and B) from Ko Rawi, Tarutao National Park, Satun Province (C, E and F) from Hala-Bala Wildlife Sanctuary, Narathiwat Province (D) from Ton Te Waterfall nature trail, Khao Banthat Wildlife Sanctuary, Trang Province. Photos: Chatchai Ngernsaengsaruay.

Figure 4 Korthalsia laciniosa.

(A–C) Leaves and leaflets. (D–F) Leaf and leaflets on both surfaces, glaucous below. (A, C–E) from Ban Tamot Forest Protection Unit, Khao Banthat Wildlife Sanctuary, Phatthalung Province (B, F) from Ko Rawi, Tarutao National Park, Satun Province. Photos: Chatchai Ngernsaengsaruay.

Distinguishing characteristics. Korthalsia laciniosa is recognised as a robust high-climbing rattan; leaves with 6–15 leaflets on each side of rachis; leaflets variable in shape and size, rhomboid, sometimes narrowly or broadly rhomboid, 15–40 × 5.5–24.8 cm [length/width ratio (1.2–)1.5–3(–4.4)]; leaflets glaucous on abaxial surface; green leaf sheaths and covered with caducous reddish-dark brown and black scales; ocreas tightly sheathing, closely sheathing at first but tending to dissolve into a loosely fibrous net-like tube; leaf sheaths and ocreas with flattened triangular spines.

Distribution, ecology and conservation status. Among the Thai species, Korthalsia laciniosa is the most widespread. It is distributed from the Andaman and Nicobar Islands, southern Myanmar, and Indo-China to the Philippines (Fig. 2B). The species is known from numerous localities and has a large EOO of 5,699,101.83 km2 and an AOO of 372 km2. In Thailand, it is naturally distributed in the south-eastern region (Chanthaburi and Trat provinces) and the peninsular region (Chumphon, Surat Thani, Krabi, Nakhon Si Thammarat, Phatthalung, Trang, Satun, Songkhla, Yala, and Narathiwat provinces). Within Thailand, the species has an EOO of 210,902.45 km2 and a small AOO of 96 km2. It typically occurs in lowland tropical evergreen rainforests, at elevations ranging from near sea level to 550 m a.s.l. Due to its very wide distribution, presence in numerous localities, and the absence of any known imminent threats to its populations or habitats, we propose to assign this species a conservation status of LC.

In addition to the distribution of this species reported in the Flora of Thailand (Barfod & Dransfield, 2013), we here document its natural occurrence in Trat, Surat Thani, Krabi, Nakhon Si Thammarat, Trang, Satun, Songkhla, and Yala provinces.

Vernacular names. Wai kadao (หวายกะเดา) (Surat Thani, from the specimen A. F. G. Kerr 13231); Wai dao (หวายเดา) (Trat, from the specimen C. Phengklai et al. 13488; Phatthalung, field observation); Wai dao yai (หวายเดาใหญ่) (Narathiwat, from the specimen I. Vongkaluang 492); Wai daeng (หวายแดง) (Nakhon Si Thammarat, from the specimen P. Suvarnkoses 639); Wai sadao (หวายสะเดา) (Trat, from the specimen A. F. G. Kerr 6933; Chumphon, from the specimen A. F. G. Kerr 11511; Satun, from the specimen A. F. G. Kerr 14082); Wai sadao nam (หวายสะเดาน้ำ) (Narathiwat); Wai sadao yai (หวายสะเดาใหญ่) (Peninsular); Cha-wa (จะวะ) (Khmer-Chanthaburi, from the specimen T. Smitinand 3481); Da-nae-ka-yo (ดาแนกาเยาะ) (Malay-Narathiwat).

Uses. The Urak Lawoi, an ethnic minority group in Ko Tarutao national park, Satun province, use the stems of Korthalsia laciniosa to make fish-traps, known locally as “Sai (ไซ)”. Indigenous people living in Phatthalung and Trang provinces use the stems of this species to make bird cages, while the young shoots are traditionally infused in alcohol to enhance vitality and physical strength. In the past, the stems of this rattan were also used to build the frameworks of boat roofs, especially for traditional wooden boats, but this practice is not done today. Over the past 15–20 years in Ko Kut, Trat province, the stems of Korthalsia laciniosa have been harvested for sale and local use. Barfod & Dransfield (2013) reported that this species produces a coarse cane.

Notes. Calamosagus laciniosus was described by Griffith (1845), based on material collected from “Mergui Archipelago”. He did not assign a collector number, designate a holotype, nor indicate the herbarium where the material was deposited. The species was subsequently transferred to Korthalsia by Von Martius (1845). Dransfield (1981) selected the material at BR [without barcode] as the holotype, with isolectotypes at FI [without barcode] and K [without barcode]. However, in accordance with Art. 9.6 of the ICN (Turland et al., 2018), these are syntypes, and Dransfield’s action constitutes a first-step lectotypification. We were unable to locate the lectotype at BR and the isolectotype at FI but we did locate an isolectotype at K [K000697940].

Calamosagus wallichiifolius was described by Griffith (1845), based on material collected from “Malacca, Kussan” without a collector number. Although Griffith did not designate a holotype or indicate the herbarium in which the specimen was deposited, the single gathering he cited qualifies as the holotype in accordance with Art. 9.5 of the ICN (Turland et al., 2018). The species was later transferred to the genus Korthalsia by H. Wendland (de Kerchove de Denterghem, 1878). Dransfield (1981) cited a specimen at BR [without barcode] as the holotype, without designating any isotypes, which is consistent with this interpretation. Despite extensive searches, we have not located the lectotype at BR.

Korthalsia teysmannii was described by Miquel (1861), based on a specimen collected by Teysmann from Muara Dua (originally published as “Muaraduwa”), Palembang, Sumatra. Miquel did not designate a holotype or indicate the herbarium where the specimen was housed, nor did he provide a collector number. Under Art. 9.6 of the ICN (Turland et al., 2018), the original material should be treated as syntypes. Dransfield (1981) subsequently selected the specimen at BO [without barcode] as the lectotype, with an isolectotype at FI [without barcode], constituting a first-step lectotypification under Art. 9.17 of the ICN (Turland et al., 2018). Therefore, the name K. teysmannii has been lectotypified by Dransfield (1981). We have not located the lectotype at BO, but we have found an isolectotype at FI [FI086076].

Korthalsia grandis was described by Ridley (1907), based on specimens collected in Singapore, specifically from Seletar (originally spelled “Selitar” in the protologue) and Bukit Mandai. Furtado (1951) designated the Ridley collection from Seletar (without collector number) as the lectotype, without mentioning any isolectotypes. We have located the lectotype at SING [SING0016962].

The specimens C. Niyomdham & W. Ueachirakan 1804 and C. Niyomdham & W. Ueachirakan 1805 at BKF, collected from Pa Ye, Su-ngai Padi district, Narathiwat province, along the margin of a peat swamp forest, were originally identified as Korthalsia tenuissima Becc. and K. grandis, respectively. However, both specimens belong to K. laciniosa. Notably, this species has not previously been recorded from peat swamp forest habitats. Korthalsia tenuissima is known only in Peninsular Malaysia, while K. grandis is now considered a synonym of K. laciniosa.

As noted by Barfod & Dransfield (2013), Korthalsia laciniosa occurs in lowland and hill evergreen forests. Based on our observations, the species is found in lowland tropical evergreen rainforests at elevations ranging from near sea level to 550 m a.s.l.

Specimens examined. Thailand. South-Eastern: Chanthaburi [Khlong Lamphong, Pong Nam Ron district, with infl., 3 Sep 1956, T. Smitinand 3481 (BKF); Soi Dao district, with infl. (Leaves and inflorescences belong to Korthalsia laciniosa and fruits belong to another rattan genus), 9 Jun 1963, K. Larsen 9942 (BKF); Khao Soi Dao wildlife sanctuary, sterile, 17 May 2013, S. Tagane et al. T1527 (BKF)]. Trat [Khlong Mayom, Ko Chang, sterile, 6 Apr 1923, A. F. G. Kerr 6933 (AAU, BK, K [K000667723], L [L4191309, L4191310]); Khlong Ao Salat, Ko Kut, sterile, 8 Apr 2002 (as K. cf. grandis), C. Phengklai et al. 13488 (BKF); Makha Yak nature trail, Ko Kut subdistrict, Ko Kut district, 11° 39′ 33″ N 102° 34′ 21″ E, 200 m, sterile, 11 Dec 2024, C. Ngernsaengsaruay et al. Kl01-11112024 (BKF); Huang Nam Khiao nature trail, Ko Kut subdistrict, Ko Kut district, 11° 39′ 07″ N 102° 34′ 55″ E, 160 m, sterile, 11 Dec 2024, C. Ngernsaengsaruay et al. Kl02-11112024 (BKF); Roadside near Good Time Resort, Ko Kut subdistrict, Ko Kut district, 11° 40′ 33″ N 102° 33′ 07″ E, 120 m, sterile, 12 Dec 2024, C. Ngernsaengsaruay et al. Kl03-12112024 (BKF)]. Peninsular: Chumphon [Khao Thong (originally “Kao Tong” on the label), infr. with young fr., 18 Jan 1927, A. F. G. Kerr 11511 (AAU, BK, K [K000667727], L [L4191268], NY [NY02320575]); unreadable locality, Lang Suan district, sterile, 23 Feb 1927, A. F. G. Kerr 12101 (BK, K [K000667724])]. Surat Thani [Khao Nong (originally “Kao Nawng” on the label), with infl., 9 Apr 1927, A. F. G. Kerr 13231 (AAU, BK, K [K000667729])]. Krabi [Khao Pra Bang Khram wildlife sanctuary, with infl., 5 Oct 1996 (as K. grandis), A. S. Barfod et al. 43825 (BKF, K [K000667728]); Forest Resource Management Office No. 12, Krabi Branch, Thap Prik subdistrict, Mueang Krabi district, sterile, N. Mianmit, 23 May 2025, personal observation with photos]. Nakhon Si Thammarat [Khao Luang, sterile, 28 Jul 1953, P. Suvarnkoses 639 (BKF); Ban Khiriwong, Kamlon subdistrict, Lan Saka district, sterile, 26 Jul (year not recorded), Unknown s.n. (BKF5357); Ban Nai Mong, Phrom Lok subdistrict, Phrom Lok district, sterile, N. Mianmit, 8 May 2025, personal observation with photos]. Phatthalung [Ban Tamot, infr. with young fr., 20 Mar 1979 (as Korthalsia sp.), J. Dransfield & C. Boonnab JD5440 (K [K000667735]); Ban Rai Tok, Ban Tamot forest protection unit, Khao Banthat, Tamot subdistrict, Tamot district, 07° 16′ 09″ N 100° 01′ 23″ E, 130 m, sterile, 5 May 2025, Ngernsaengsaruay et al. Kl08-05052025 (BKF), ibid., sterile, 5 May 2025, C. Ngernsaengsaruay et al. personal observation with photos; Ban Tamot forest protection unit, Khao Banthat wildlife sanctuary, Tamot subdistrict, Tamot district, 07° 15′ 17″ N 100° 02′ 41″ E, 70 m, sterile, 5 May 2025, Ngernsaengsaruay et al. Kl09-05052025 (BKF)]. Trang [Ton Te waterfall, Palian district, sterile, 14 Nov 1990, A. S. Barfod & W. Ueachirakan 141344 (AAU); Khao Chong, Chong subdistrict, Na Yong district, 07° 32′ 58″ N 99° 47′ 01″ E, 70 m elevation, sterile, 3 May 2025, C. Ngernsaengsaruay et al. personal observation with photos; Ton Te Waterfall nature trail, Ton Te forest protection unit, Khao Banthat wildlife sanctuary, Palian subdistrict, Palian district, 07° 17′ 47″ N 99° 53′ 04″ E, 140 m, sterile, 4 May 2025, Ngernsaengsaruay et al. Kl07-04052025 (BKF); ibid., 07° 17′ 51″ N 99° 53′ 05″ E, 150 m elevation, sterile, 4 May 2025, C. Ngernsaengsaruay et al. personal observation with photos]. Satun [Ko Adang, infr. with young fr., 14 Jan 1928, A. F. G. Kerr 14082 (AAU, BK, K [K000667734]); Ko Tarutao national park, with infl., A. F. G. Kerr 14164 (AAU, K [K000667725], L [L4191266]); Ko Rawi, Tarutao national park, Ko Sarai subdistrict, Mueang Satun district, 06° 33′ 24″ N 99° 15′ 27″ E, 10 m, sterile, 20 Apr 2025, Ngernsaengsaruay et al. Kl06-20042025 (BKF)]. Songkhla [Boriphat waterfall, Rattaphum district, sterile, 19 Oct 1991 (as K. grandis), K. Larsen et al. 42409 (AAU)]. Yala [Labu mine near Bannang Sata district, sterile, 19 Nov 1972 (as Korthalsia sp.), T. Whitmore et al. TLW3126 (BKF, K [K000667722]); Sukthalai waterfall, Taling Chan subdistrict, Bannang Sata district, with infl., 20 Mar 2025, N. Mianmit personal observation with photos]. Narathiwat [Pa Ye, Su-ngai Padi district, sterile, 14 Apr 1988 (as K. tenuissima), C. Niyomdham & W. Ueachirakan 1804 (BKF); ibid., sterile, 14 Apr 1988 (as K. grandis), C. Niyomdham & W. Ueachirakan 1805 (BKF); Locality unspecified, sterile, s.d., I. Vongkaluang 492 (K [K000667733]); Locality unspecified, sterile, s.d., I. Vongkaluang 490 (K [K000667736]); Locality unspecified, sterile, IDRC Rattan Project 1982–1985, Kasetsart University (as K. rigida), I. Vongkaluang 491 (K [K000667739]); Hala-Bala wildlife sanctuary, Phu Khao Thong subdistrict, Sukhirin district, 05° 48′ 39″ N 101° 48′ 04″ E, 440 m, sterile, 8 Feb 2025, Ngernsaengsaruay et al. Kl04-08022025 (BKF); ibid., 05° 48′ 15″ N 101° 47′ 14″ E, 540 m, sterile, 8 Feb 2025, Ngernsaengsaruay et al. Kl05-08022025 (BKF)].

3. Korthalsia rigida Blume, Rumphia 2: 167. 1843 (Blume, 1843). Type: Indonesia, Sumatra, Palembang, s.d., Praetorius s.n. (lectotype, designated by Dransfield (1981), L [without barcode] not seen; isolectotype FI [without barcode] not seen).

= Korthalsia polystachya Mart., Hist. Nat. Palm. 3: 210, t. 172. 1845. ≡ Calamosagus ochriger Griff., Palms. Brit. E. India 31, t. 216. 1850. Type: as for K. polystachya Mart. Type: Peninsular Malaysia; Malacca, Fernandez s.n. (lectotype, designated by Dransfield (1981), BR [without barcode] not seen; isolectotypes FI [without barcode] not seen, K [without barcode] not seen).

= Korthalsia ferox Beec. var. malayana Becc. in Hook. f., Fl. Brit. India 6: 476. 1893. Type: Peninsular Malaysia, Perak, Larut, Sep. 1884, H. Kunstler 6563 (lectotype, designated by Dransfield (1981), CAL [without barcode] not seen; isolectotype K digital image! [K000207577]).

= Korthalsia hallieriana Becc., Ann. Roy. Bot. Gard. (Calcutta) 12(3): 142. 1918. Type: Indonesia, Borneo, West Kalimantan, Archip. Ind. Soengai Kenepai, 1893–1894, H. (J.) G. Hallier 2135 (lectotype, designated by Dransfield (1981), FI digital image! [FI014349]; isolectotype BO [without barcode] not seen).

= Korthalsia paludosa Furtado, Gard. Bull. Singapore 13: 313. 1951. Type: Peninsular Malay, Johore, Sungai Kayu, 7 Mar 1937, Kiah 32344 (holotype SING [without barcode] not seen; isotype digital image! K [K000667675]).

Non-myrmecophilous, slender to moderate-sized, clustered, high-climbing rattan, up to 50 m or more long, basally and aerially branching. Stems with leaf sheaths 1.5–3 cm diam.; internodes 20–30 cm long. Leaves 1.18–2.40 m long including petiole and cirrus; leaflets 5–7 on each side of rachis, regularly arranged and rather distant; leaf sheath dull green, densely covered with persistent whitish-grey indumentum and caducous blackish-brown and black scales, and armed with scattered single, paired or grouped and flattened triangular or bulbous-based spines, each spine 4.5–14 mm long, 2–8.5 mm wide at the base; ocrea 3.5–7.5 cm long, dull green with a reddish-brown or reddish-dark brown margin, tightly sheathing, apex truncate with a narrow tattering margin, sometimes splitting or tending to dissolve into a fibrous net-like tube, with scales as the leaf sheath, and densely armed with single, paired or grouped and flattened triangular or bulbous-based spines, each spine 4.5–14 mm long; petiole 18–42 cm long, 5–9 mm wide and 3.5–7 mm thick, adaxially shallowly grooved or flattened proximally and flattened distally, abaxially rounded, petiole 14–40.5 cm apart from each other, sparsely armed with single, paired or grouped and recurved spines, 2–10 mm long; rachis 47–90 cm long, adaxially flattened at the lower part and angular at the upper part, armed with sparsely single, paired or grouped and recurved spines, each spine 3–9 mm long; petiole and rachis green adaxially, pale green abaxially, and with indumentum and scales as the leaf sheath; cirrus slender, green, 0.40–1.20 m long, armed with scattered short, grouped, sometimes paired or single and recurved spines. Leaflets alternate, sometimes subopposite, rhomboid or sometimes broadly rhomboid, 12–31 × 7.5–22 cm [length/width ratio (1.2–)1.5–2.8], the lowest pair of leaflets not smaller than the others, the distal margins praemorse, glossy dark green and glabrous, except for the main veins with black and reddish-dark brown scales on adaxial surface, glaucous and covered with black and reddish-dark brown scales on abaxial surface, main veins 7–14, diverging from the lamina base, raised above, transverse veinlets conspicuous above; petiolule laterally flattened, 0.5–2.6 cm long, densely covered with persistent whitish-grey indumentum and caducous blackish-brown and black scales; young leaflets pale green (Fig. 5).

Figure 5 Korthalsia rigida.

(A) Basally branching. (B) Climbing stem. (C) Stem and leaf. (D and E) Leaf sheaths, densely covered with persistent whitish-grey indumentum, and armed with scattered spines and tightly sheathing ocreas, apex truncate with a narrow tattering margin, sometimes splitting, and armed with densely spines. (F) Leaf and leaflets on adaxial surface. (G) Cut stem and leaflets on both surfaces, glaucous below. (A–G) from Hala-Bala Wildlife Sanctuary, Narathiwat Province. Photos: Chatchai Ngernsaengsaruay.

Distinguishing characteristics. Korthalsia rigida can be recognised as a moderate-sized climbing rattan; leaves with 5–7 leaflets on each side of rachis; shape of leaflets, rhomboid or sometimes broadly rhomboid, 12–31 × 7.5–22 cm [length/width ratio (1.2–)1.5–2.8]; leaflets glaucous on abaxial surface; dull green leaf sheaths and densely covered with persistent whitish-grey indumentum; ocreas tightly sheathing, dull green with a reddish-brown or reddish-dark brown margin, apex truncate with a narrow tattering margin, sometimes splitting or tending to dissolve into a fibrous net-like tube; leaf sheaths and ocreas with flattened triangular or bulbous-based spines.

Distribution, ecology and conservation status. Korthalsia rigida is widely distributed across southern Myanmar, Peninsular Thailand, Borneo, and the Philippines (Palawan) (Fig. 2C). It has been recorded from numerous localities and has a large EOO of 3,073,095.43 km2 and an AOO of 288 km2. In Thailand, the species is restricted to the peninsular region (Ranong, Trang, and Narathiwat provinces), where it has a considerably smaller EOO of 18,717.23 km2 and an AOO of 20 km2. The species occurs in lowland tropical evergreen rainforests at elevations of 70–300 m a.s.l. Given its broad distribution range, the high number of known localities, and the absence of major threats, we propose a conservation status of LC.

Vernacular names. Wai dao nu (หวายเดาหนู) (Trang); Da-nae-ti-ku (ดาแนตีกุ๊) (Malay-Narathiwat).

Uses. Barfod & Dransfield (2013) reported that this species produces a coarse cane.

Notes. Korthalsia rigida was originally described by Blume (1843), based on two gatherings from forests along rivers, both in Sumatra, whence a fragment of this palm was sent to him by his esteemed friend Prætorius and in Borneo, as attested by specimens collected by the distinguished Mr.Korthals “Habit. In sylvis circa fluvios, cum in Sumatra, unde fragmentum hujus Palme ad me misit Vir amicissimus Prætorius, tum in Borneo, quod collecta a Korthals V. Cl. Specimina testantur.” Blume did not specify the herbarium in which the specimens were deposited; therefore, following Art. 9.6 of the ICN (Turland et al., 2018), these gatherings are considered syntypes. Dransfield (1981) selected the specimen Praetorius at L [without barcode] as the lectotype, with an isolectotype at FI [without barcode]. We have not been able to locate the lectotype at L or the isolectotype at FI.

Korthalsia polystachya was described by Von Martius (1845) from the Malay Peninsula, where it is called Rotang Danam, as noted by Eman. Fernandez and Griffith: “Crescit in peninsula Malaccensi, ibidem Rotang Бапаш dicta: Еman. Fernandez, Griffith.” von Martius did not specify the herbarium in which the specimens were deposited; therefore, following Art. 9.6 of the ICN (Turland et al., 2018), these specimens are considered syntypes. Dransfield (1981) selected the specimen Fernandez s.n. from Malacca at BR [without barcode] as the lectotype, with isolectotypes at FI [without barcode] and K [without barcode]. We have not been able to locate the lectotype at BR nor the isolectotypes at FI and K.

Korthalsia ferox var. malayana was described by Beccari & Hooker (1893), based on two gatherings: “Larut, Perak (Hort. Cale. 6563)” and “Gunong, Tjick, Sept. 1844, Scortechini”. Beccari and Hooker did not specify the herbarium in which the specimens were deposited; therefore, following Art. 9.6 of the ICN (Turland et al., 2018), these gatherings are considered syntypes. Dransfield (1981) selected the specimen Kunstler 6563 at CAL [without barcode] as the lectotype, with an isolectotype at K [without barcode]. We have not been able to locate the lectotype at CAL; however, we have located an isolectotype at K [K000207577].

Korthalsia hallieriana was described by Beccari (1918), based on material collected in Dutch Borneo by Hallier during the Borneo-Expedition in 1893–1894 (No. 2135, Buitenzorg Herbarium). Following Art. 9.6 of the ICN (Turland et al., 2018), these specimens are considered syntypes. Dransfield (1981) selected the specimen at FI [without barcode] as the lectotype, with an isolectotype at BO [without barcode]. We have not been able to locate the lectotype at BO. However, we have located an isolectotype at FI [FI014349].

Korthalsia paludosa was originally described by Furtado (1951), based on Kiah 32344, collected from Sungai Kayu, Johore, Malaya, although he did not specify the herbarium in which it was deposited. Dransfield (1981) subsequently confirmed that the holotype is deposited at SING [without barcode], with an isotype at K [without barcode]. We have not been able to locate the holotype at SING. However, we have located an isotype at K [K000667675].

Korthalsia rigida is distributed in Peninsular Thailand, Peninsular Malaysia, Sumatra, Borneo, and the Philippines (Palawan) (Barfod & Dransfield, 2013). However, based on our observations of herbarium specimens, we have identified three additional collections: Hooker s.n. at K [K000667638] from the Mergui Archipelago, southern Myanmar; S. Shahimi & A. Loo 18 at K [K001193052] from MacRitchie Forest, Singapore; and S. Shahimi & A. Loo 29 at K [K001287191] from MacRitchie Reservoir, Singapore. These records represent an extension of the known distribution range and they are the first record of K. rigida in southern Myanmar and Singapore.

Korthalsia rigida is reported to occur in hill forests up to 700 m altitude (Barfod & Dransfield, 2013) but our own observations show that the species is found in lowland tropical evergreen rainforests, at elevations ranging from 70 to 300 m a.s.l.

Specimens examined. Thailand. Peninsular: Ranong [30–70 km south of Ranong, with infr., 26 Apr 1974, K. Larsen & S. S. Larsen 33397 (L [L1413028])]. Trang [Khao Chong, sterile, 19 Mar 1979, J. Dransfield & C. Boonnab 5430 (BKF, K [K000667737]); Khao Chong, with infl., 20 Oct 1993, A. B. Pedersen 45176 (AAU, BKF K [K000667738])]; Narathiwat [1,500 m nature trail, Hala-Bala wildlife research station, Hala-Bala wildlife sanctuary, Lo Chut subdistrict, Waeng district, 05° 47′ 47″ N 101° 49′ 50″ E, 90 m, sterile, 5 Feb 2025, Ngernsaengsaruay et al. Kr01-05022025 (BKF); ibid., 05° 47′ 48" N 101° 49′ 47" E, 80 m, sterile, 5 Feb 2025, Ngernsaengsaruay et al. Kr02-05022025 (BKF); ibid., 05° 47′ 45″ N 101° 49′ 51″ E, 70 m, sterile, 8 Feb 2025, Ngernsaengsaruay et al. Kr03-08022025 (BKF); Chat Warin waterfall nature trail, Budo-Su-ngai Padi national park, 8 Feb 2025, C. Ngernsaengsaruay, N. Mianmit & P. Chanton personal observation with photos].

4. Korthalsia scortechinii Becc. in Becc. & Hook. f., Fl. Brit. India 6(19): 475. 1893 (Beccari & Hooker, 1893). Type: Peninsular Malaysia, Perak, s.d., B. Scortechini s.n. (holotype FI digital image! [FI014352]).

Myrmecophilous, moderate-sized, clustered, high-climbing rattan, up to 20 m or more long. Stems including leaf sheaths 1.4–2 cm diam.; internodes 20–35 cm long, cut stems with clear jelly-like exudate. Leaves 1.55–2.58 m long including petiole and cirrus; leaflets 6–9 on each side of rachis, regularly arranged and rather distant; leaf sheath green or pale yellowish-green, almost entirely covered by the swollen ocrea, with scattered caducous, reddish-brown and black scales, and sparsely armed with short, single and flattened triangular spines, 1.5–5 mm long, 1–3.5 mm wide at the base; ocrea inflated, green, pale yellowish-green or pale yellow, turning dull pale brown when dry, ellipsoid, 9–19.5 × 5.7–10 cm, clasping the stem, coriaceous, with scales as the leaf sheath, and armed with scattered short, single flattened triangular and erect or slightly recurved spines, each spine 1.5–6 mm long; petiole 10.5–36.5 cm long, 0.5–1 cm wide and 3.5–6.5 mm thick, adaxially shallowly grooved or flattened proximally and flattened distally, abaxially rounded, petiole 14.5–40 cm apart from each other, sparsely armed with short, single or sometimes paired and recurved or sometimes erect spines, each spine 1–4 mm long; rachis 41–82 cm long, adaxially flattened proximally and angular distally, sparsely armed with short, single, paired or grouped and recurved spines, each spine 1.5–5.5 mm long; petiole and rachis green adaxially, pale green or pale yellowish-green abaxially, and with scales as the leaf sheath; cirrus slender, green, 0.65–1.64 m long, armed with scattered short, grouped, sometimes paired or single and recurved spines. Leaflets alternate, sometimes subopposite or opposite, narrowly rhomboid or rarely rhomboid, 16–37.5 × 3–13.5 cm [length/width ratio (2.6–)3–6.3], the lowest pair of leaflets often smaller than the others, the distal margins praemorse, glossy dark green and glabrous on adaxial surface, glaucous and covered with black and reddish-dark brown scales on abaxial surface, main veins 5–8, diverging from the lamina base, raised above, transverse veinlets conspicuous above; petiolule absent; young leaflets pale green (Fig. 6).

Figure 6 Korthalsia scortechinii.

(A–C) Habitats and habit. (D) Clear jelly-like exudate secreted from cut stem. (E and F) Inflated ocreas and forming an ant chamber. (G) Leaf and leaflets on abaxial surface. (H) Cut stem, showing leaf sheaths and inflated ocreas, and leaflets on both surfaces. (A–H) from Hala-Bala Wildlife Sanctuary, Narathiwat Province. Photos: Chatchai Ngernsaengsaruay.

Distinguishing characteristics. Korthalsia scortechinii can be recognised as a myrmecophilous, moderately sized climbing rattan; large ellipsoid, inflated ocreas (9–19.5 × 5.7–10 cm) that clasp the stem, with scattered short, flattened triangular and erect or recurved spines and inhibited by usually abundant black fierce ants within the ocrea chamber; mostly narrowly rhomboid and sessile leaflets (length/width ratio 3–6.3); leaflets glaucous on abaxial surface; stem size (with leaf sheaths 1.4–2 cm diam); and with clear jelly-like exudate secreted from cut stems.

Distribution, ecology and conservation status. Korthalsia scortechinii has a relatively narrow distribution from Peninsular Thailand to Peninsular Malaysia (Fig. 2D). Despite its restricted range, the species is known from many localities, with an EOO of 62,596.22 km2 and a small AOO of 40 km2. In Thailand, it is known only from Narathiwat province, where it has a small EOO of 140.20 km2 and a relatively small AOO of 16 km2. The species inhabits lowland tropical evergreen rainforests at elevations of 80–550 m a.s.l. Given its high number of known localities and the absence of significant threats, we propose the conservation status of LC.

Vernacular names. Wai kung (หวายกุ้ง) (from the specimen C. Niyomdham 5633), Wai dao lek (หวายเดาเล็ก), Wai sadao lek (หวายสะเดาเล็ก) (Narathiwat); Ro-tae-u-dae (รอแตอูแด) (Malay-Narathiwat).

Uses. In Narathiwat province, the stems of Korthalsia scortechinii are used in combination with bamboo to make the frameworks of fish- and shrimp-traps. The fish-traps are locally known as “Bubu” or “Bubo” in the Malay-Narathiwat dialect. The stems of this species can be used to make handles for pillow-beating tools and rope holders for tying monkeys. They are also used as supplementary materials in furniture making (e.g., crafting rattan armchairs, crafting rattan beds), weaving, combined with other rattan species, but they are not part of the main structure (Fig. 7).

Figure 7 Utilization of Korthalsia scortechinii in Thailand.

(A) A supplementary material in furniture making. (B–D) Crafting rattan armchairs and crafting rattan bed, weaving, combined with other rattan species. (E) A rope holder for tying monkeys. (F) A handle for pillow-beating tools. (G) Fish-traps (left and right) and a shrimp-trap (middle). (H) Fish-traps, locally known as “Bubu” or “Bubo” in the Malay-Narathiwat dialect. (I–L) Fish-traps. (A–D) from Sukhirin District, Narathiwat Province (E–L) from Waeng District, Narathiwat Province. Photos: Chatchai Ngernsaengsaruay (A–F, H–L); Nittaya Mianmit (G).

Notes. Korthalsia scortechinii was described by Beccari & Hooker (1893), based on specimen collected at “Perak” by Scortechini. Although Beccari and Hooker did not explicitly designate a holotype or indicate the herbarium in which the specimen was housed, the single gathering cited qualifies as the holotype, in accordance with Art. 9.5 of the ICN (Turland et al., 2018). Dransfield (1981) referred to a specimen at FI [without barcode] as the holotype, consistent with this interpretation. We have located the holotype at FI, represented by digital image FI014352.

Korthalsia scortechinii is found in hill dipterocarp forest up to 700 m altitude (Barfod & Dransfield, 2013). Our observations indicate that this species also occurs in lowland tropical evergreen rainforests at elevations of 80–550 m a.s.l.

Specimens examined. Thailand. Narathiwat [Hala-Bala wildlife sanctuary, sterile, 10 Feb 1996, C. Niyomdham 4566 (BKF); ibid., with infl., 29 Oct 1998, C. Niyomdham 5633 (BKF); Si Sakhon district, infr. with fr., 7 Mar 2001, C. Niyomdham & P. Puudjaa 6446 (BKF, K [K000667741]); Locality unspecified, sterile, IDRC Rattan Project 1982–1985, Kasetsart University, I. Vongkaluang 500 (K [K000667740]); 1,500 m nature trail, Hala-Bala wildlife research station, Hala-Bala wildlife sanctuary, Lo Chut subdistrict, Waeng district, 05° 47′ 43″ N 101° 49′ 41″ E, 80 m, sterile, 5 Feb 2025, Ngernsaengsaruay et al. Ks01-05022025 (BKF); ibid., 05° 47′ 41″ N 101° 49′ 41″ E, sterile, 5 Feb 2025, Ngernsaengsaruay et al. Ks02-05022025 (BKF); Hala-Bala wildlife sanctuary, Phu Khao Thong subdistrict, Sukhirin district, 05° 48′ 04″ N 101° 46′ 26″ E, 400 m, sterile, 7 Feb 2025, Ngernsaengsaruay et al. Ks03-07022025 (BKF)].

Discussion

Hodel & Vatcharakorn (1998) listed six species of Korthalsia that were recorded in Thailand: Korthalsia ferox Becc., K. flagellaris, K. laciniosa, K. rigida, K. rostrata, and K. scortechinii but our research found no specimens of K. ferox and K. rostrata from Thailand, which agrees with the findings of Barfod & Dransfield (2013). The native range of K. ferox is restricted to Borneo, while K. rostrata is known from Peninsular Malaysia, Singapore, Sumatra, and Borneo, but not Thailand. Therefore, this study recognises only four species of Korthalsia as occurring in Thailand, i.e., K. flagellaris, K. laciniosa, K. rigida, and K. scortechinii.

In this study, the leaflets of Korthalsia flagellaris are very narrowly rhomboid or sometimes narrowly rhomboid, 18.5–37 × 1.5–5 cm. They are glossy dark green and glabrous on the adaxial surface, while the abaxial surface is paler, not glaucous, and covered with densely black and reddish-brown scales proximally, and densely reddish-brown scales distally. These findings differ somewhat from previous descriptions. Barfod & Dransfield (2013) described the leaflets as lanceolate, up to 30 × 4.5 cm, dark bluish-green above, brownish beneath, with frequent bands of deciduous chocolate-colored scales. Hodel & Vatcharakorn (1998) noted them as narrowly rhomboid, about 30 × 4–5 cm, with a rusty silver-brown lower surface. The variation observed in shape, size range, surface coloration, and scale characteristics may reflect intraspecific variation, environmental influences, or differences in developmental stage. Notably, this study highlights a more consistent presence of dense reddish-brown to black scales and a narrower leaflet shape than those reported in previous studies.

In this study, the leaf sheaths of Korthalsia laciniosa are described as green, sometimes pale yellowish-green, covered with caducous reddish-dark brown and black scales, and sparsely armed with single or sometimes paired and flattened triangular spines. These observations show some variation from previous descriptions. Barfod & Dransfield (2013) reported the leaf sheaths as bright green, with abundant reddish-brown indumentum and chocolate-colored scales, and sparsely armed with triangular spines. Hodel & Vatcharakorn (1998) described them as tubular, green, and either unarmed or sparsely armed with triangular spines. The differences in sheath coloration and armature observed in this study, particularly the presence of caducous dark scales and paired flattened spines, may reflect regional variation, intraspecific morphological plasticity, or different developmental stages of the plants examined.

Conclusions

Korthalsia is the sole genus within the subtribe Korthalsiinae and is highly distinctive among rattans due to its unique morphology. It exhibits a remarkable combination of features, including climbing, hapaxanthic, hermaphroditic habits; aerially branching stems; once-pinnately compound leaves with rhomboid (diamond-shaped) leaflets with distal margins praemorse; and inflorescences with cylindrical, catkin-like rachillae.

Our investigations of Korthalsia in Thailand, provides and updated vegetative morphological descriptions and an identification key for the four species: K. flagellaris, K. laciniosa, K. rigida, and K. scortechinii. In Thai Korthalsia species, vegetative characteristics offer reliable diagnostic traits for species identification.

Three Thai Korthalsia species, K. flagellaris, K. scortechinii, and K. rigida are confined to the peninsular region; K. flagellaris to peat swamp forests in Narathiwat province, K. scortechinii, a myrmecophilous species, which is also found only in Narathiwat, occurs in lowland tropical evergreen rainforests, and K. rigida which has a broader distribution covering Ranong, Trang, and Narathiwat provinces, where it is found in lowland tropical evergreen rainforests. In contrast, K. laciniosa the most widespread species of the genus, occurs in both the south-eastern and peninsular regions of Thailand, where it is also found in lowland tropical evergreen rainforests. All Thai Korthalsia species are currently assessed as LC under the IUCN conservation status.

Main uses of Korthalsia in Thailand are for producing fishing gear and local handicrafts; young shoots are also eaten or used in traditional medicine.

We would like to thank the curators and staff of the herbaria BK and BKF for their assistance during our visits and for granting access to the herbarium specimens. We also extend our gratitude to the institutions that maintain virtual herbarium databases, including A (and GH), AAU, BR, E, GH, K, L (and U), P, SING, US, and MICH, MO, MW, and NY, accessed via the Global Biodiversity Information Facility (GBIF). We are grateful to all plant collectors who contributed specimens of the rattan genus Korthalsia. Special thanks are extended to Sunate Karapan, Chief of the Hala-Bala wildlife research station, Narathiwat province, to Mongkon Daengkun, Chief of the Sirindhorn Peat Swamp Forest Nature Research and Study Center, Narathiwat province, to Nakain Kaveethanathum, Assistant Chief of Tarutao national park, and to Chutipong Ponlawat, Chief of the Khao Banthat wildlife sanctuary, along with their staff, for their generous support and facilitation during the fieldwork. We also wish to thank Weereesa Boonthasak for her kind assistance in the laboratory.

Additional Information and Declarations

Competing Interests

The authors declare that there are no competing interests.

Author Contributions

Chatchai Ngernsaengsaruay conceived and designed the experiments, performed the experiments, analyzed the data, prepared figures and/or tables, authored or reviewed drafts of the article, and approved the final draft.

Nittaya Mianmit performed the experiments, analyzed the data, prepared figures and/or tables, authored or reviewed drafts of the article, and approved the final draft.

Nisa Leksungnoen performed the experiments, authored or reviewed drafts of the article, and approved the final draft.

Phruet Racharak performed the experiments, authored or reviewed drafts of the article, and approved the final draft.

Suwimon Uthairatsamee performed the experiments, authored or reviewed drafts of the article, and approved the final draft.

Pichet Chanton performed the experiments, prepared figures and/or tables, authored or reviewed drafts of the article, and approved the final draft.

Tushar Andriyas performed the experiments, authored or reviewed drafts of the article, and approved the final draft.

Wirongrong Duangjai performed the experiments, authored or reviewed drafts of the article, and approved the final draft.

Field Study Permissions

The following information was supplied relating to field study approvals (i.e., approving body and any reference numbers):

We obtained permission to collect specimens from the Department of National Parks, Wildlife and Plant Conservation, Ministry of Natural Resources and Environment, MNRE 0910.5803/7512.

Data Availability

The following information was supplied regarding data availability:

Raw data is available in Table 1.

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
