# Peer review of "An account of Korthalsia (Arecaceae) rattans and their uses in Thailand"

_PeerJ, doi:10.7717/peerj.19935_

## Round 0.1 · original submission · Major Revisions

I agree with the two reviewers in several comments such as that the manuscript should be edited by a fluent English speaker, and that the types of the species should be included, and that a table should be eliminated. One of the reviewers included a detailed review attached here. Please explain carefully how you dealt with every suggestion by the two reviewers.

**Language Note:** The review process has identified that the English language must be improved. PeerJ can provide language editing services - please contact us at [email protected] for pricing (be sure to provide your manuscript number and title). Alternatively, you should make your own arrangements to improve the language quality and provide details in your response letter. – PeerJ Staff

Reviewer 1 ·

Basic reporting

.

Experimental design

.

Validity of the findings

.

Additional comments

attached

Annotated reviews are not available for download in order to protect the identity of reviewers who chose to remain anonymous.

Reviewer 2 ·

Basic reporting

Generally, the paper is generally well written and structured. The paper presents an introduction to the genus Korthalsia with an emphasis on species located in Thailand. The structure follows a conventional scientific format, beginning with taxonomic background and progressing to distribution and uses. However, the organization within sections would benefit from improvement, as information about taxonomy, morphology, and distribution is occasionally intermingled without clear transitions. The manuscript demonstrates adequate English proficiency but contains several grammatical errors.

Experimental design

Their research relies on the study of herbarium (physical and virtual) and collected specimens from the field. The authors provide figures for all studied species. Too summarize information on biogeography, and species distribution the authors also provide a map. All the findings are summarized in figures and tables.

Validity of the findings

The findings are an updated from the previous study with more comparative information on vegetative morphology, distribution, ecology, conservation status and utilization of Thai Korthalsia species.

Additional comments

The abstract is not well written. Generally an abstract is a summary of ideas, work and methods while offering insights and highlight on key findings. In this sense, the abstract should begin with an issue to solve, touch on the objective and continue to highlight on procedures or steps taken to create a solution. Then mention about 2-3 findings which are highly significant to the article and end the abstract with soft notes of concluding remarks while also offering one recommendation.

The introduction section should be about awareness creation. The first paragraph has some developments but the components related to definitions or justifications (personal views on authors) should be removed.

Avoid using the statement Thai Korthalsia. Use other term or statement to mention the species location in Thailand.

Abstract:
L26-29: Delete
L30-34: ‘Vegetative features can be ………. and characteristics’. Delete this statement.
L40: Add word ‘Lastly,’ before K. laciniosa is the most ….
L43-46: Summarize the utilization Korthalsia species in Thailand.

Keywords:
Use single words which are general and not already used in the title. Keywords are special or unique words to identify the article when it is placed with other articles in the same clustering field. Use general words so that persons who type these words could be able to retrieve your work, read and cite it.

Introduction:
The introduction relies heavily on older references (Dransfield, 1981) for utilization information. Consider incorporating more recent literature on economic importance and conservation.
L60-63: This sentence needs to restructuring
L63-64: Delete.
L75-77: Cannot find the information from the cited references given. Also, the citation format for the POWO (2025) is different from the website.
L95: Narathiwat Province not Narathiwat Provinces. Change all.
L104: “Finding” should be plural

Material and Method:
L112. Delete full stop
L119: Dransfield et al. 2004

Result
L134: Delete word ‘In’
L140-142: This statement is partially true. Some species from genus Calamus also demonstrate similar branching pattern.
L144-145: Delete ‘… of Korthalsia species…’
L182-183: Delete statement.
L186: Delete.
L187-188: ‘leaflets rhomboid….. broadly rhomboid’. Delete
Distinguishing characteristics: Most the information in this section taken from Description part. Avoid repetitive statement.

Discussion
L534-542: This sentence needs to restructuring. Summarize the findings.

References
Arrange in alphabetical order

---

## Round 0.2 · Minor Revisions

Please consider every suggestion by Reviewer 1 which will improve your article. Some suggestions were in the attached document and others directly here.

Reviewer 1 ·

Basic reporting

.

Experimental design

.

Validity of the findings

.

Additional comments

This is dramatically improved compared to version 1. I have read the docx version submitted and made many editorial changes, mostly to improve the English language, but also others. All my comments to the main text can be found in track changes on the enclosed docx version of the manuscript [20250620 HBA comments peerj-118204-(1)_PeerJ-research-manuscript_Korthalsia_in_Thailand_clean_version_20250524]

The docx version did NOT include the tables and figures, so I could not mark those sections. But they were included in the pdf version, so I have the following general comments:

Table 1 lists herbarium data related to the collections made by the authors during fieldwork. That same data is NOT included in the specimen citation after the description of each species, as they should be. So, please take all information from Table 1 and insert it under specimen citations after the species descriptions.

Table 2 is a tabulation of the same information which is given under the descriptions of each of the four species. So, it is redundant and the whole Table 2 should be deleted. Or alternatively Table 2 can be maintained and the data included can then be deleted from the descriptions of the species.

Table 3 is a tabulation of information about the uses of all Korthalsia species, also such ones occurring outside of Thailand. But the table is not explained or discussed in the text. Since it is a very valuable Table, the authors should add a section about used o Korthalsia in general in the introduction to the paper.

Table 4 specifies all discrepancies between the current understanding of Korthalsia and previous studies. This is obviously an excellent working tool while preparing the manuscript, but all these intermediate notes do not belong in the final paper – so Table 4 should be deleted.

Figures 1–7 are beautifully prepared and illustrates the text very well. If desired I would be happy to edit the text, but I would like to do that in track-changes on a docx version.

Annotated reviews are not available for download in order to protect the identity of reviewers who chose to remain anonymous.

---

## Round 0.3 · accepted · Accept

I am glad that you addressed all the comments by the reviewer in the second round of reviews. The paper has improved very much. My only concern is that Figure 2, the distribution map, seems to lack contrast between land and sea. However, when reviewing the proofs, you can check this issue.